# Persistent Infection with Rotavirus Vaccine Strain in Severe Combined Immunodeficiency (SCID) Child: Is Rotavirus Vaccination in SCID Children a Janus Face?

**DOI:** 10.3390/vaccines7040185

**Published:** 2019-11-16

**Authors:** Maria Antonia De Francesco, Giovanni Ianiro, Marina Monini, Cesare Vezzoli, Richard Fabian Schumacher, Silvia Giliani, Giovanni Lorenzin, Francesca Gurrieri, Arnaldo Caruso

**Affiliations:** 1Institute of Microbiology, Department of Molecular and Translational Medicine, University of Brescia-Spedali Civili, 25123 Brescia, Italy; giovanni.lorenzin@hotmail.it (G.L.); francescagurrieri@me.com (F.G.); arnaldo.caruso@unibs.it (A.C.); 2Department of Food Safety, Nutrition and Veterinary Public Health, Istituto Superiore di Sanità, 00161 Rome, Italy; giovanni.ianiro@iss.it (G.I.); marina.monini@iss.it (M.M.); 3Pediatric Intensive Care Unit, Children’s Hospital, ASST-Spedali Civili, 25123 Brescia, Italy; cesare.vezzoli@asst-spedalicivili.it; 4Haemato-oncology and BMT Unit, Children’s Hospital, ASST-Spedali Civili, 25123 Brescia, Italy; fabian.schumacher@unibs.it; 5A. Nocivelli Institute for Molecular Medicine, Department of Molecular and Translational Medicine, University of Brescia, 25123 Brescia, Italy; silvia.giliani@unibs.it

**Keywords:** SCID, vaccination, rotavirus, transplantation

## Abstract

We report the first case, to our knowledge, in Italy, of a severe combined immunodeficiency patient with a persistent rotavirus infection due to a vaccine derived strain. Rotavirus was detected by enzyme immunoassays and RT-PCR in stool specimens for five months. The persistent infection was resolved after complete immune reconstitution achieved by hematopoietic stem cell transplantation. This case underlines the importance of neonatal SCID_screening.

## 1. Case

Severe combined immunodeficiency (SCID) includes inherited diseases with an impairment of T cell development, often associated with profound defects of B and/or NK cell differentiation. These genetic defects lead to severe and repeated infections by opportunistic microorganisms, to autoimmunity and neoplasies [1]. Clinical treatment is necessary to prevent severe life-threatening diseases and is based on hematopoietic stem cell transplantation or gene therapy when available [2].

Rotavirus infection is responsible for severe acute gastroenteritis and for half a million deaths a year, in children aged <5 years, especially in low and middle-income countries. For this reason, in 2009, the World Health Organization (WHO) recommended a rotavirus vaccination in all national immunization programs. Currently two live oral vaccines are licensed in Italy: RotaTeq, composed of five various human–bovine reasserting rotavirus strains (MercK) and Rotarix, a monovalent vaccine (GlaxoSmithKline). Rotavirus vaccination has been strongly recommended since 2017 for all infants between the age of six and twelve weeks (first dose).

Infants with severe combined immunodeficiency, have severe side effects after the administration of live vaccines such as BCG (Bacillus Calmette-Guérin), measles, mumps, rubella (MMR) and varicella [3].

Also, for the rotavirus vaccination, a general concern is being raised about its safety in SCID patients, who develop severe clinical symptoms that require hospitalization. This was witnessed by different cases reported first in the United States [4,5,6] and in Australia [7], then in Europe (Germany and UK) [8,9], and more recently also in Japan [10]. This is one of the reasons, programmes of neonatal screening for SCID were recently introduced nation-wide in the USA [11] and just a year ago also in the Italian region of Tuscany.

Our patient was a full-term Italian female infant who was born without complications and to unrelated parents. According to the Italian routine childhood vaccination schedule, she received the first dose of Rotarix at 3 months of age and the second dose a month later. Though no immediate adverse effects were observed, she developed persistent severe diarrhoea and was hospitalized 10 weeks later for fever, failure to thrive, and severe hypotonia. A few days later, after severe lymphopenia and a life-threatening pneumonia triggered a suspicion of SCID, she was transferred to the paediatric intensive care in our hospital.

Bronchoalveolar lavage was positive for *Pneumocystis jiroveci* which was successfully treated with intravenous high-dose trimethoprim-sulfamethoxazole. Rotavirus was detected in stool specimen by enzymatic immunoassay, but no other enteric viruses or pathogenic bacteria were found. Diarrhoea and positive rotavirus stool specimens lasted for five months until the age of 11 months.

Given the severe lymphopenia (absolute lymphocyte count 80 cells/mm^3^), subpopulations were analyzed and the results are shown in the Table 1. Immunoglobulin levels were below the detection limits. TRECS and T cell proliferative response to mitogens were completely absent completing the picture of T-B-NK+SCID (Table 1) Next generation sequencing allowed to detect the homozygous mutation c.G2210A:p.R737H in the recombination activating gene 1 (RAG1).

To characterize this rotavirus infection, we investigated ten stool samples collected at different time intervals during the next five months.

Total viral RNA was extracted using the QIAmp Viral RNA Mini Kit (Qiagen, Monza Italy) from a 10% (w/v) faecal suspension, following the manufacturers’ instructions. Viral RNA was subjected to reverse transcriptase PCR (RT-PCR) of genes 9 and 4, encoding the outer capsid protein (VP7) and the viral hemagglutinin (VP4), respectively. The amplification was performed by using the forward primer Beg9 (5′-GGCTTTAAAAGAGAGAATTTCCGTCTGG-3′) and the reverse primer End9 (5′-GGTCACATCATACAATTCTAATCTAAG-3′) for VP7, and by using the forward primer Con3 (5′-TGGCTTCGCCATTTTATAGACA-3′) and the reverse primer Con2 (5′-ATTTCGGACCATTTATAACC-3′) for VP4 [12,13]. Genotyping of VP7 (G-type) and VP4 (P-[type]) was performed following European standardized protocols [14], and revealed the G1P [8] genotype for 9/10 samples. The last sample collected was rotavirus negative in both the immune-enzymatic screening test and the PCR test (genotyping and sequencing), underlining the viral clearance of the patient. The RT-PCR amplicons of VP7 and VP4 (VP8* hypervariable region) were subjected to Sanger nucleotide sequencing, revealing the highest nucleotide sequence identities (nt id.) with the vaccine strain RVA/Vaccine/USA/Rotarix-A41CB052A/1988/G1P [8] (included in the Rotarix vaccine composition) for both VP7 (nt id. Ranging between 99.46% and 99.58%) and VP4 (nt id. Ranging between 99.36% and 99.52%). On the other hand, low nucleotide sequence identities were observed with respect to wild type G1P [8] strains circulating in Italy and with respect to the rotavirus strains included in the Rotateq vaccine composition. Sequences obtained were submitted to GenBank under the following accession numbers: MN549964 to MN549981.

The deduced amino acid sequences revealed four substitutions for VP7 (E73K, E149G/A, M202T, and N238S) with respect to the Rotarix viral variant. Two out of four substitutions (positions 149 and 238) were located in VP7 antigenic sites [15].

Also, for VP4, four amino acid substitutions were observed (T73A, Y152S, F167L, and P234S) with respect to Rotarix. None of the substitutions were included in any viral antigenic epitope [16].

At 10 months of age, the patient underwent transplant of bone marrow from a matched unrelated donor after reduced-intensity conditioning (fludarabine, busulfan and anti-thymocyte globulin). She received graft-versus-host disease prophylaxis with Cyclosporine and Mycophenolate Mofetil. Rotavirus could be detected in stool specimen for another 3 weeks thereafter; however, it cleared a week later after successful T-cell engraftment (300 CD3+ cells). Shortly thereafter, the patient was discharged from the transplant unit in good general conditions.

This case is the first reported in Italy. It reinforces the concept that live rotavirus vaccination in SCID patients can cause severe clinical symptoms such as diarrhoea, weight loss and a long viral persistence as demonstrated by the detection of rotavirus in multiple stool samples over 5 months; this suggests a continuous viral replication with a total clearance obtained only after successful immune reconstitution.

## 2. Conclusions

Due to the fact that the timing of the first dose of rotavirus vaccination is scheduled in children aged between six weeks and three months (it might trigger intussusception thereafter) and congenital immune deficiency is often still undiagnosed at that time, we suggest general newborn screening should be introduced as soon as possible. However, in its absence, clinicians should be particularly careful about clinical symptoms potentially related to SCID, such as failure to thrive and persistent or recurrent infections, in order to avoid the administration of rotavirus vaccine in this particular category of patients. Furthermore, persistent rotavirus infection in a vaccinated child should always trigger an immunologic work-up.

Therefore, while rotavirus vaccination constitutes a risk for infants born with SCID, it is safe for the general population (a different side of the same coin—‘a Janus face’) preventing more than 80% of severe cases of rotavirus diarrhea in countries with low death rates [17].

## Figures and Tables

**Table 1 vaccines-07-00185-t001:** Immunological characteristics of the severe combined immunodeficiency (SCID) patient.

Absolute Cell Counts	80 Cells/mL
Lymphocyte subpopulations (age corrected normal % range)	%
CD3+ T cells (52–83)	1.9
CD4+ T cells (31–58)	1
CD8+T cells (16–40)	0.9
CD19+ B cells (5–18)	0
CD16+56+NK cells (5–27)	94
Immunophenotype	T-B-NK+
Gene defect	RAG1
Serum Immunoglobulins (age corrected normal range)	g/L
IgG (270–1100)	<35
IgM (20–170)	<5.25
IgA (110–115)	<7.83
T cell proliferation	PHA-

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
