# Peer review of "Persistent Infection with Rotavirus Vaccine Strain in Severe Combined Immunodeficiency (SCID) Child: Is Rotavirus Vaccination in SCID Children a Janus Face?"

_vaccines, 2019, doi:10.3390/vaccines7040185_

Round 1

Reviewer 1 Report

Rotarix (manufactured by GlaxoSmithKline) is a live-attenuated monovalent vaccine against rotavirus. In this case study, the authors report, for the first time in Italy, an infant with severe combined immunodeficiency (SCID) that developed persistent rotavirus infection after vaccination with Rotarix. Although this finding is not novel (other reports cited in this manuscript include children in the United States, Australia, Germany, the United Kingdom, and Japan), this is the first case reported in Italy and this phenomenon has not been reported in the Vaccines journal before, to the best of my knowledge. There are several minor modifications and grammatical corrections that would improve this manuscript, listed below.

Minor modifications:
1. Please add a short introductory paragraph giving a brief description of rotavirus and SCID. Some information from the abstract can potentially be moved to this section. The addition of an introductory paragraph would greatly help with manuscript flow.

2. If Tables are allowed for Case Reports, the analysis of cell populations and Ig levels would be more clear in tabular format.

3. According to the Instructions for Authors, manuscripts involving human subjects, tissues, or data must declare that informed consent was given. The authors did include this statement. They also must include a project identification code, date of approval, and name of the ethic committee or institutional review board (IRB) in the manuscript. Please include the reference number for the ethical approval or IRB approval, if any was given, in the manuscript.

4. Lines 47-51: Please include reference for RNA extraction, PCR protocol, and primers used for PCR.

Grammatical Changes:
1. Lines 16-18: Please change to "Currently, two live oral rotavirus vaccines are licensed in Italy: RotaTeq, composed of five various human-bovine reassortant rotavirus strains (Merck), and Rotarix, a monovalent vaccine (GlaxoSmithKline). Rotavirus vaccination..."

2. Line 20: Please change to "responsible for".

3. Line 33: Please add a period at end of sentence.

4. Line 46: Please delete the misplaced parentheses at end of sentence.

5. Line 50: Please cite website according to Author Instructions.

6. Line 51: Please change to "The last sample collected was rotavirus negative.

7. Lines 55-56: Please change to "...(nt id. Ranging between 99.36% and 99.52%). On the other hand, low nucleotide sequence identities..."

8. Line 72: Please change to "...a general concern is being raised about its safety..."

9. Line 76: Please change to "...recently introduced nation-wide in the USA..."

10. Line 78: Please change to "...reinforces the concept that live rotavirus vaccination..."

11.Lines 80-81: Please change to "...samples over 5 months; this suggests continuous viral replication..."

12. Lines 86-87: Please change to "However, in its absence, clinicians should..."

13. Line 87 and 88: Please add comma after "related to SCID" (line 87) and after "recurrent infections" (line 88).

14. Line 89: Please change "Rotavirus" to "rotavirus" to be consistent with the rest of the manuscript.

Author Response

Dear Editor and Referees

Thank you very much for giving us the opportunity to revise our paper

Answers to referee 1:

An introductory paragraph has been inserted in the case section and it has been deleted from the Abstract. A Table has been inserted with the analysis of cell populations and Ig levels. Ethic statement was already present at the end of the paper. Due to the case report nature of the manuscript, no ethical approval was required. References for RNA extraction, PCR protocol, and primers used for PCR have been inserted.

All grammatical errors evidenced by the referee have been corrected.

Please refer to the attachment for detailed reply.

Reviewer 2 Report

The authors present a case of vaccine-derived, persistent rotavirus infection in an infant with SCID. While novel to Italy, vaccine-derived rotavirus infection in SCID infants has been globally reported including in Europe. A case is made for physicians to screen newborns for SCID, or in the absence of screening, watch for signs of SCID prior to administering live attenuated vaccines.

The title could be improved to better reflect the topic of the paper: 1)the “Janus face” reference is not well addressed in the report, 2) the case report is on a single child, not children, 3) the vaccine-derived, persistent rotavirus infection is not addressed in the title.

The abstract mostly contains information that would better be address in a background section. A brief summary of the case and conclusions be presented in the abstract. An introduction section that briefly covers rotavirus vaccination, SCID diagnosis, and previously described diagnosis and treatment of vaccine-derived rotavirus in SCID patients is needed.

Lines 65-68: Details on the protocol for bone marrow transplant, timeline, and confirmatory lab tests for viral clearance should be provided.

Lines 70-71 do not need to be a separate paragraph.  While vaccine-derived viral infections following live attenuated vaccination are commonly known, when mentioning specific vaccines or instances, citations would be beneficial. This material would be more appropriate in an introduction section.

The authors should address in the discussion or conclusion that the benefit of rotavirus vaccination greatly outweighs the risk of vaccine-derived, persistent viral infection both due to the rarity of SCID and the efficacy and safety of live attenuated rotavirus vaccines.

There are numerous grammatical errors throughout the report. Professional editing would greatly improve readability.

Author Response

Dear Editor and Referees

Thank you very much for giving us the opportunity to revise our paper

Answers to referee 2

The title has been modified according to the referee’s suggestions. The term ‘Janus face’ has been addressed in the Conclusions section The abstract has been modified according to the referee’s suggestions. An introduction section has been added Details on the protocol for bone marrow transplant, timeline, and confirmatory lab tests for viral clearance have been inserted Parts of this section have been moved to the Introduction section and the paragraph has been not separated. We have added the importance of rotavirus vaccination in general population in the Conclusions section. The paper has been revised by an English mother tongue reader

Please refer to the attachment for detailed reply.
